# Vascular Calcification in Rodent Models—Keeping Track with an Extented Method Assortment

**DOI:** 10.3390/biology10060459

**Published:** 2021-05-22

**Authors:** Jaqueline Herrmann, Manasa Reddy Gummi, Mengdi Xia, Markus van der Giet, Markus Tölle, Mirjam Schuchardt

**Affiliations:** 1Department of Nephrology and Medical Intensive Care, Charité—Universitätsmedizin Berlin, Corporate Member of Freie Universität Berlin and Humboldt-Universität zu Berlin, Hindenburgdamm 30, 12203 Berlin, Germany; Jaqueline.Herrmann@charite.de (J.H.); manasa.gummi@charite.de (M.R.G.); mengdi.xia@charite.de (M.X.); Markus.vanderGiet@charite.de (M.v.d.G.); markus.toelle@charite.de (M.T.); 2Department of Chemistry, Biochemistry and Pharmacy, Freie Universität Berlin, Königin-Luise-Straße 2+4, 14195 Berlin, Germany

**Keywords:** 3R, vascular calcification, imaging, biomarkers

## Abstract

**Simple Summary:**

Arterial vessel diseases are the leading cause of death in the elderly and their accelerated pathogenesis is responsible for premature death in patients with chronic renal failure. Since no functioning therapy concepts exist so far, the identification of the main signaling pathways is of current research interest. To develop therapeutic concepts, different experimental rodent models are needed, which should be subject to the 3R principle of Russel and Burch: “Replace, Reduce and Refine”. This review aims to summarize the current available experimental rodent models for studying vascular calcification and their quantification methods.

**Abstract:**

Vascular calcification is a multifaceted disease and a significant contributor to cardiovascular morbidity and mortality. The calcification deposits in the vessel wall can vary in size and localization. Various pathophysiological pathways may be involved in disease progression. With respect to the calcification diversity, a great number of research models and detection methods have been established in basic research, relying mostly on rodent models. The aim of this review is to provide an overview of the currently available rodent models and quantification methods for vascular calcification, emphasizing animal burden and assessing prospects to use available methods in a way to address the 3R principles of Russel and Burch: “Replace, Reduce and Refine”.

## 1. Introduction

Vascular calcification as a major pathophysiological contributor to cardiovascular morbidity and mortality has been extensively investigated in basic and clinical research. However, research is hampered by the fact that vascular calcification is not a single entity but is rather heterogeneous, for example in terms of calcification localization (intimal or medial, anatomical site of the vessel), calcification size (micro- and macrocalcification), and the biochemical pathways involved in the formation of ectopic calcification [1]. Although many participating pathways have been described in animal models, the transfer to humans and the development of therapeutics with a clinical benefit is challenging. Among other factors, the species shift from rodents to humans is hampered by the fact that rodents are not prone to the development of vascular calcification. Therefore, extensive manipulations are necessary to induce vascular calcification in rodents that at least in part resemble the human condition within a manageable experimental time frame.

Several reviews have summarized the available animal models used in vascular calcification research [2,3,4]. However, the extent of the animal burden connected with each model, either by harmful genetic modification or by the harm of the treatment procedure to induce the desired disease grade, varies across the different models. Moreover, several quantification methods that vary in specificity and sensitivity have currently been established for the assessment of vascular calcification with different impacts on animal numbers and burden.

This review aims to summarize and evaluate the different methods for identifying and quantifying vascular calcification in rodents, especially in mice. The choice of a suitable experimental setup, including the selection of adequate research models and an appropriate identification and quantification methodology, are decisive in ensuring reproducibility and transferability. Besides, this allows the transformation of the 3R principles formulated by Russell and Burch in 1959: “Replace, Reduce and Refine” [5] from theory into practice.

## 2. Rodent Models for Induction of Vascular Calcification

As the rodent models for studying vascular calcification have been reviewed recently [2], only a brief overview is given. The aim of Table 1 is rather to point out the differences between the models in the expected animal burden.

Briefly, in rodents, vascular calcification is induced by genetic modification, e.g., knockout of endogenous inhibitors of ectopic calcification, induction of chronic kidney disease (CKD) by operation or feeding a special diet (e.g., adenine) and is rarely naturally occurring. The expected calcification differs regarding the localization among different vessels and soft tissues and among the vessel layers. Localization, for example in the intima, predominately takes place in models of altered lipoprotein systems like ApoE^−/−^ mice, while CKD mice models often exhibit a predominant calcification of the media. Furthermore, calcification size differs from microcalcifications; for example, in the early stages of the disease, to macrocalcifications. Therefore, the effect strength with subsequent assay sensitivity affects the necessary sample sizes required for statistical analysis. Longitudinal research approaches using the same animal as a control can reduce statistical variances and thus reduce the sample size required. Furthermore, the choice of the model can contribute to refinement, e.g., avoiding operative burdens by CKD induction via an adenine diet instead of nephrectomy. However, the uremia-related symptoms such as body weight loss and skin itching leads to a high animal burden.

Next to the application of in vivo models, ex vivo and in vitro settings offer valuable alternatives. Primary cells, e.g., vascular smooth muscle cells (VSMC), can be derived from animals after sacrificing by explanted outgrowth or a digestion method [6] or are commercially available, for instance from human donors. VSMC cell lines, such as MOVAS or A7r4, are commercially available. Although in vitro models are a valuable alternative to address the 3R principle, their use is limited due to the loss of tissue organization and missing interactions with the extracellular matrix, as well as systemic interactions [2].

Different ex vivo settings, e.g., the isolated perfused artery [7], might at least partially overcome some drawbacks of in vitro settings and can further help to reduce the animal burden. The application of substances in a dedicated ex vivo system avoids in vivo application systems associated with a recurring animal burden, such as drug pumps, oral gavage or injections. As part of refinement strategies, the application of ex vivo and in vitro methods can help to reduce the burden for animals caused by experimental procedures and can help to reduce the number of animals required for experimental testing.
biology-10-00459-t001_Table 1Table 1Introduction of vascular calcification in various rodent models (modified after [2]).Method of InductionModelAspects of Model/Indication for Animal Burden Naturally occurringDBA2 [8]‑Female mice are more prone to vessel calcification.CY+ rat with autosomal dominant PKD [9,10,11]‑Fatal in homozygous animals within 3–4 weeks, heterozygous Cy/+ rats develop progressive PKD and uremia, with males being more affected.‑Animals aged 38 weeks on normal phosphorous diet develop vascular calcification of the thoracic aorta with medial localization.‑Animals often present tremulous, lethargic, and extremely ill.LPK disease rat [12,13]‑Animals suffer from cyst development (3 weeks), hypertension (6 weeks), renal dysfunction (3 months), progressing to end-stage renal disease (5 months).‑LPK rats have increased aortic stiffness and increased aortic calcification.OperationKidney reduction (electrocautery, nephrectomy) [14,15]‑Animals suffer from surgery related stress, pain, anesthesia, surgical wounds, and analgesia; surgical-associated death rate.‑Variability in calcification progression.‑High burden of progressive renal failure with the possibility of acute renal failure.‑Uremia-associated changes in eating and drinking behavior with weight loss and skin itching.Feeding/Substance applicationAdenine [16,17,18,19,20,21,22,23,24,25,26]‑High burden of progressive renal failure with the possibility of acute renal failure.‑Uremia-associated changes in eating and drinking behavior with weight loss and skin itching.‑Reduced food intake due to altered taste with resulted weight loss.‑Refinement in relation to operation-induced induction of renal insufficiency (no surgical wounds, anesthesia).Vitamin D [27,28,29,30,31,32,33,34]‑Used as diet-component (often in combination with others) to increase progression of calcification.‑Acute hypercalcemia, hyperphosphatemia, blood pressure increase, and weight loss.Phosphate [27,28,29,30,31,32,33,34,35,36,37,38,39,40,41]‑Used as a diet-component (often in combination with others) to increase progression of calcification; concentration differs between protocols.Streptozotocin [42] ‑Glycemia, reduced food consumption and body weight loss.Cholesterol Rich Chow [43]‑Used as a diet-component to increase atherosclerosis and associated plaque calcification.‑Development of atherosclerotic lesions with macrophage infiltration.PCSK9-AAV [44,45] ‑Mice receiving an injection of PCSK9 and high fat feeding for 15 to 20 weeks develop calcification comparable to Ldlr^−/−^ mice, without adverse responses reported in association with the injection. ‑An advantage of this method is the reduction in necessary animal numbers for breeding, thus offering an opportunity for reduction of animal numbers.Genetic modificationKlotho^−/−^ [46,47]‑After normal development for 3–4 weeks, mice exhibit growth retardation, inactiveness, condition decline and death at 8–9 weeks of age, mice suffer from abnormal walking pattern, osteopenia, emphysema and ectopic calcification, including media calcification of the aorta.Fgf-23^−/−^ [48,49]‑Marked growth retardation after 2 weeks in Fgf-23^−/−^ mice with a reduced life span of a maximum of 13 weeks.‑Mice exhibit hyperphosphatemia, increased serum levels of calcium and vitamin D, vascular calcification of the kidney, abnormal bone development, and uncoordinated movement.Galnt^−/−^ [50]‑Mice have normal life expectancy with slower growth compared to wildtype mice, with infertility of homozygous males, but not females.‑Mice have increased serum phosphate and serum calcium levels, but not ectopic calcification.Tcal/Tcal [51]‑Mice have hyperphosphatemia and ectopic calcification, e.g., at the aorta and kidney, with males being infertile and females being fertile.‑No shortened life expectancy or growth retardation reported.Abcc6^−/−^ [52,53] ‑Abcc6^−/−^ mice have no reduced lifespan, nor increased mortality or gross abnormalities compared to wildtype mice, they spontaneously develop vessel calcification.Enpp1 (Enpp^−/−^, Enpp^ttw/ttw^, Enpp1^asj/asj^) [54,55,56,57,58]‑Enpp1^−/−^ develop abnormal calcification of the aorta and have reduced weight and body length.‑Enpp^ttw/ttw^ have calcification, e.g., of kidney and aorta and when challenged with overload of phosphate exhibit signs of premature aging and die within 3 weeks after phosphate overload following an inactive and marantic episode.‑Enpp^asj/asj^ have stiffened joints, reduction in physical activity and weight loss during aging; high phosphorous diet worsened calcification and drastically shortened life span to approximately 6 weeks and mice suffer from calcification of aorta and coronary arteries.Lmna [59,60,61,62,63] ‑Lmna^L530P/L530P^ mice develop growth retardation 4–6 days after birth and die within 4–5 weeks.‑Lmna^G609G/G609G^ are infertile but appear healthy until 3 weeks of age, upon which they show growth retardation, curvature of the spine, abnormal posture, and premature death within 3–5 months.‑Homozygous G608G mice have growth retardation and declined activity after seven months followed by premature death and exhibit large vessel calcification, while heterozygous G608G model exhibits cardiovascular phenotype but lacks other symptoms of Progeria.Fetuin-A^−/−^ [64,65]‑Fetuin-A^−/−^ mice on a 129Sv/C57BL/6 background appear healthy with normal lifespan and ectopic calcification being detectable in several female ex breeders, with ectopic calcification being increased by mineral/vitamin D–rich feeding.‑Fetuin-A deficiency on DBA2 background results in reduced viability and fertility and mice displayed severe systemic calcification.Opg^−/−^ [66]‑Opg^−/−^ mice exhibit vascular calcification of the aorta and renal arteries. They have decreased mineral bone density with osteoporosis worsening with age and reduced viability, likely attributed to bone fractures.Mgp^−/−^ [67]‑Mgp^−/−^ exhibit severe phenotype changes in comparison to wildtype mice after 2 weeks of age: they become shorter, their heart rate increases, and they die within 2 months following rupture of the aorta. Animals exhibit severe calcification, e.g., of the media of the aorta and of the coronary arteries.Opn^−/−^ [68,69]‑Opn^−/−^ mice appear healthy, are fertile and exhibit no obvious abnormalities, in particular no vascular calcifications. ‑In mice deficient in Mpg and Opn, life expectancy is further reduced when compared to Mgp^−/−^ Opn^+/+^ mice. Cause of death is haemorrhage as consequence of vascular calcification at 4 weeks of age.Madh6^−/−^ [70]‑Adult Madh6^−/−^ appear to be in good health, but a substantial share of Madh6^−/−^ do not survive until post weaning, making Madh6^−/−^ mice underrepresented in the progeny of heterozygous intercrosses. Madh6^−/−^ aged six weeks or older show calcification of the outflow tract of the heart.ApoE^−/−^ [71,72]‑Mice appear healthy with no reproductive problems. ApoE^−/−^ mice have significantly increased cholesterol levels, reduced HDL levels and develop plaque calcification on normal chow.Ldlr^−/−^ [73,74]‑Mice deficient in Ldlr exhibit increased plasma cholesterol but overall mild pathologic manifestations and are fertile while on normal chow. Mice are diet hyper responsive when fed a cholesterol-enriched diet and develop atherosclerosis on high fat diet.ApoE3 Leiden [75,76]‑Higher serum cholesterol and triglyceride level (hyperlipidemia).‑Development of atherosclerotic lesions with macrophage infiltration.Abbreviation: Abcc6—ATP binding cassette subfamily C member 6; ApoE—apolipoprotein E; asj—ages with stiffened joints; Enpp1—ectonucleotide pyrophosphatase phosphodiesterase; Fgf-23—fibroblast growth factor 23; Galnt—alNAc trans-ferase; HDL—high density lipoprotein; Lldlr—low density lipoprotein receptor; Lmna—gene encoding the lamin A/C protein; LPK—lewis polycystic kidney; Madh6—mothers against decapentaplegic homolog 6; Mgp—matrix Gla protein; Opg—osteoprotegerin; Opn—osteopontin; PCSK9-AAV—proprotein convertase subtilisin/Kexin type 9 - adeno-associated virus vector; PKD—polycystic kidney disease; Tcal—tumoural calcinosis; ttw—tip toe walking.

## 3. Methods for the Detection of Vascular Calcification in Rodents

A variety of different methods for detecting vascular calcification are available, both for detecting the deposition of calcium, phosphate, and hydroxyapatite, and for the detection of surrogate parameters like vessel stiffness and blood biomarkers. With the concurrent development of specialized preclinical imaging technologies suitable for small rodent imaging, the application of novel in vivo imaging is becoming increasingly available. Core facilities often provide the extensive hardware equipment and trained personal for in vivo technologies. With the availability of in vivo imaging techniques such as micro-computed tomography (µCT), positron emission tomography (PET), intravital microscopy (IVM) and a combination of these techniques, longitudinal studies monitoring calcification progression become realizable. Longitudinal studies offer a huge potential for reducing animal numbers. To avoid an animal sacrifice for every single read out, fewer animal numbers can be required for statistical testing due to reduced statistical variance. Longitudinal studies can also facilitate finding optimal treatment times by offering multiple assessments of effect strength. Nevertheless, next to these advantages, an additional burden for animals by repetitive testing, such as repeated anesthesia and operations to assess the imaging side, have to be taken into account. Table 2 sums up the available methods for detecting vascular calcification in rodents, ordered according to the underlying technology and their potential for longitudinal analysis. To date, the majority of in vivo imaging experiments were at least in part proof-of-concept studies. Animals were often sacrificed after in vivo measurements for a subsequent validation of results with more established standard laboratory procedures, such as histological staining. Nevertheless, results from these studies indicate that longitudinal studies employing new in vivo methods are becoming increasingly available. 

### 3.1. Biochemical Markers for Calcification

Vascular calcification is an organized process involving multiple inhibitory and promoting pathways. Describing and evaluating all of these established pathways that can regulate vascular calcification is beyond the scope of this review. Nevertheless, selected pathways for vascular calcification and markers of VSMC osteoblastic trans-differentiation have been the topic of numerous recent reviews [2,77,78,79]. 

A variety of calcification-related proteins are expressed locally, intracellularly and secretory and are used for the prognostic description of calcified vessels. While the detection of biomarkers in animal fluids, e.g., blood, provides a unique opportunity for longitudinal research, the correlation of assessed biomarkers with hard endpoints is often disputable. Some of the biomarkers, like serum calcium, were correlated with vascular calcification or other endpoints in large human studies [78]. 

To describe the tissue changes during vessel calcification, resident biomarkers are detected. The assembly of protein–protein interactions by Song et al. evaluates the relationship between different endo-phenotypes of VSMC and demonstrates that the calcifying cell phenotype overlaps with the endophenotype of inflammation, fibrosis and thrombosis [80]. Some of these markers are currently used for the description of cellular changes during the calcification progression. 

#### 3.1.1. Blood Biomarkers of Vascular Calcification

Table 3 summarizes the currently investigated biomarkers for vascular calcification studies. The biomarkers Fgf-23 and klotho are interconnected: klotho is a necessary co-factor for binding of Fgf-23 to its receptor. Some of these markers can also be imaged with techniques presented later in this review. For example, Alexa-Flour-labeled Fetuin-A was used in combination with IVM [81]. 

#### 3.1.2. Resident Biomarkers of Vascular Calcification Localized in Tissue

Calcification is mostly monitored by the detection of osteoblastic trans-differentiation markers including bone morphogenetic protein-2 (Bmp-2) [87], runt-related transcription factor 2 (Runx2, also known as core-binding factor alpha 1 (Cbfa1)) [87,88], tissue non-specific alkaline phosphatase (Alp) [89], msh homeobox (Msx) [90] as well as smooth muscle-specific markers, e.g., smooth muscle 22-kDa protein (Sm22-α) [88] and alpha smooth muscle actin (α-Sma) [88]. 

Bmp-2 is a key mediator of vascular calcification and is associated with local induction of mineralization and inflammation, promoting osteogenic and cartilaginous differentiation of pluripotent mesenchymal progenitor cells [91]. It was found that Bmp-2 is expressed by various cells in atherosclerotic lesions [92] and induces expression on Runx2 and Msx [93]. Several studies have correlated some common mediators of endothelial dysfunction with increased Bmp-2 expression and calcification [94]. 

Runx2 is a crucial regulator of osteoblast differentiation. Just as it is essential in bone formation, the expression of Runx2 in VSMC is an early definite marker of osteoblast differentiation. As known from CKD patients, Runx2 is selectively expressed in calcified arterial tissue [95]. Different factors, including, but not limited to high glucose, uremia, oxidative and genotoxic stress [39,87,96,97], that contribute to vascular calcification, can induce the expression of Runx2 [98]. 

Msx was shown to have calcifying properties both in vitro and in vivo and was found to act via canonical WNT signaling [90,99].

Alp is a functional phenotypic marker of osteoblasts, and its activity is also commonly used as a molecular marker of vascular calcification. Studies believe that Alp activity is critical to the formation of hydroxyapatite during endochondral ossification by hydrolyzing pyrophosphate, and therefore providing phosphate necessary for hydroxyapatite formation by likewise reducing the crystallization inhibiting PPi [100]. When expressed in the vasculature, Alp is believed to have a similar effect [84]. Therefore, calcifying VSMC express higher levels of Alp [101], and classic atherosclerotic vascular calcification stimuli (including Bmp-2 and oxidized LDL) will also increase the Alp activity in VSMC in vitro [102,103].

Additionally, in the process of arterial calcification, the expression of VSMC contractile phenotype markers, such as α-SMA and SM22-α, are downregulated [102,104].

In conclusion, biomarkers describe the calcification progression in cells and tissue and could be used to analyze the signaling pathways involved. While blood biomarkers have the potential for longitudinal approaches, the resident biomarkers are analyzed post mortem and thus can only reflect one time point instead of disease progression. 

### 3.2. Functional Markers of Vessel Stiffness 

Stiffening of the vessel wall is a complex process involving several cellular responses as endothelial nitric oxide production, alterations in the extracellular matrix, and pathophysiologic changes of VSMC as calcification [1]. A positive correlation for vascular calcification and vessel stiffening has already been shown for humans and animals [1].

Vascular calcification is associated with advanced remodeling processes in the vessel wall and disruption of the media, causing arterial wall stiffening. Subsequently, fundamental changes in hemodynamics, including impaired *Windkessel* function and increased blood flow pulsatility, occur. 

In humans, vessel stiffness is measured via pulse wave velocity (PWV), ankle-brachial index (ABI), and pulse pressure (PP). These markers are well established in clinical practice for humans; however, the direct prediction of vascular calcification is not easy for these methods [1]. ABI as an index of peripheral arterial disease is also an indirect measure of arterial stiffness and is currently the preferred setting in clinical practice for humans. Research shows that low ABI is associated with atherosclerosis and vascular calcification in large arteries, while high ABI with medial calcification in peripheral and distal arteries [105].

The measurement of PWV and PP are also used in rodent models and provide the possibility for a longitudinal follow-up, therefore addressing 3R, even though, as already mentioned for humans, the direct prediction of vascular calcification is tentative and needs validation by other methods. Wire myography is a functional test used in experimental ex vivo settings for the analysis of vessel texture.

#### 3.2.1. Pulse Wave Velocity

PWV, the speed by which the pulse wave travels in a conduit vessel, has been used to indicate vascular stiffness related to the vascular mineralization progression. PWV is being increasingly investigated in rat and mice models [106,107]. With the continuous improvement of the technical system and the upgrading of evaluation methods, PWV might offer a simple way to assess the functional properties of different arteries in vascular calcification [108]. 

#### 3.2.2. Pulse Pressure

The mean arterial pressure (MAP) of adults does not change much, but due to the increased arterial stiffness driven by the structural and/or functional alterations of the vessel wall, PP may increase significantly [109]. Kevin et al. showed the importance of PP as a robust and essential correlate of vascular calcification, especially abdominal aortic calcification. PP has been shown to have a tendency to promote the progression of vascular calcification, which can predict the change of vascular wall calcification over time [110]. 

In Abcc6^−/−^ mice, which were found to have a slight increase in arterial stiffness and vascular calcification, PP is significantly lower compared with control mice [52].

#### 3.2.3. Wire Myography

Wire myography is an in vitro technique that allows the monitoring of the functional responses and the vascular reactivity of small vessel segments. Different vessel beds from various species, in a variety of pathological disease states, can be investigated. Changes in the feasibility of dilatation provide information on endothelial dysfunction and analysis of further alterations in vascular reactivity is feasible. Moreover, the biomechanical and passive properties of the vessel, such as the vessel diameter, elasticity, and vascular compliance, can be investigated [111,112]. After vessel dissection and cleaning from the surrounding tissue, small vessel rings are clamped in a myograph chamber with an isometric technique. The investigated drugs are applied to the buffer in the measuring chamber. Standardized experimental conditions allow the examination of pharmacological differences between the vessels [112]. 

Kirsch et al. used wire myography as a functional assay to validate the histological and chemical analysis of uremia-induced vessel mineralization in mice. They found a reduced maximum contraction in mice fed a high-phosphate diet compared to the control animals [113]. Similar results were found in a rat model of adenine-induced renal failure, where the rate of aortic relaxation was reduced in CKD rats compared to control animals [114]. The contraction rate was also reduced in mesenteric arteries and aorta in the CKD group compared to control rats [114]. Although in this study no vascular calcification could be detected, possibly because CKD induction for less than ten weeks by adenine diet with gradually reduced adenine concentration, it shows that the aorta is affected and the rate of relaxation is reduced [114]. In an Abcc6^−/−^ mouse model, wire myography was used to describe the vessel contraction potential, which was not altered, although calcium deposits and osteogenic trans-differentiation occurred in VSMC [52].

In conclusion, functional biomarkers for the description of vessel stiffness are widely used in the clinical situation. Up to now, only a few studies have used this for animal models. As mentioned above, a precise prediction of vascular calcification from functional markers is not easily given. However, a longitudinal approach is possible and therefore might help to get further information of vessel function in animals and potential changes under the treatment procedure, thus helping to find the right time point and renders the possibility for reducing the animal number.

### 3.3. Quantification and Imaging of Calcification

A variety of methods are available for the direct detection, imaging and quantification of vascular calcification. The available methods include, but are not limited to, biochemical analysis, histological staining, IVM, magnetic resonance imaging (MRI), CT, and PET, summarized in Table 2. These methods are discriminable amongst others in term of their ability to accurately localize and image the calcification, by a method’s specificity, sensitivity and robustness, as well as the destructive impact on the sample material, the eligibility for in vivo testing, the technical equipment necessary and the expenditure of work required. 

#### 3.3.1. Biochemical 

Calcium ions and *o*-cresolphthalein form a stable complex on absorbing light at approximately 575 nm that is easily quantifiable by colorimetric measurement [115]. Due to the easy protocol and low technical requirements, the *o*-cresolphthalein method is a widespread method for quantifying calcium. However, the approach has several drawbacks: the calcium has to be dissolved from tissue or cells by acidic digestion, thus destroying tissue integrity and hampering the applicability of further downstream analysis. Moreover, with this method, the identification of the calcium origin (intra- or extracellularly) is not possible, and localization of calcification is only possible to a small extent, e.g., when the tissue is sectioned and different sections are analyzed independently. The differentiation between micro- and macrocalcification or between medial or intimal calcification is also not possible. 

#### 3.3.2. Histological Staining for Calcium Deposits

Histology is a well-established standard lab technology and therefore, to date, is considered the gold standard. Staining for calcium like von Kossa and alizarin red have been established for decades [116,117]. To increase sensitivity, fluorescence dyes were established for the detection of calcium deposits in cells and tissue.

Classical histology often follows a standard workflow: tissue is dissected and either fixed, e.g., in paraformaldehyde or formalin, embedded, e.g., in paraffin, and sectioned; or alternatively, tissue is frozen after dissection and cryosectioned. Tissue sections of usually 4–6 µm thickness are then stained with the selected dye, imaged in an appropriate microscope and (semi-)quantification can then be conducted with dedicated software. Depending on the workflow, automatization is possible. The workflow and other characteristics of classical histology are summarized in Figure 1.

Due to the relatively inexpensive hardware requirements and the broad application of histology, many laboratories have equipment and histological procedures that are often included in routine lab education, thus making histology widely accessible. However, classical histology suffers from several drawbacks such as low specificity of dyes and high working expenditure for analysis of serial cuts of tissue or whole tissue sections. As the calcification foci could be spotty distributed, especially when only microcalcifications are present in the tissue, several tissue sections should be analyzed to avoid measuring bias. Extensive tissue processing can also result in post mortem modifications of morphology, including the drop out of hydroxyapatite crystals due to sectioning. 

##### Von Kossa Staining

The method of von Kossa staining is widely used to detect the presence of abnormal calcium deposits in cell cultures and histological sections. It is based on a precipitation reaction where calcium ions, bound to phosphates, are replaced by silver ions from silver nitrate. Under a light source, silver phosphate undergoes a photochemical degradation and is thereby visualized as metallic silver deposits. If counterstained with nuclear fast red, the nuclei of the cells are colored in red and the cytoplasm in pink [118]. If counterstained with H&E staining, calcium appears in a deep-blue purple [117]. While commonly used for calcification assessment, von Kossa staining is somewhat limited by its reduced specificity for calcium crystals [117].

##### Alizarin Red S Staining

For decades, alizarin red S staining has been used to identify calcium deposits in cells and tissue sections. It reacts with calcium via its sulfonate and hydroxyl groups forming bright red deposits in aqueous solution requiring a pH of at least pH 4 [119]. This reaction is not strictly specific to calcium. Other cations like magnesium or manganese may interfere, but these elements normally do not occur in sufficient concentrations to interfere with the staining [116]. Nonetheless, it should be taken into account that this stain also detects calcium-binding proteins and proteoglycans without discriminating for the presence of hydroxyapatite [120].

##### Fluorescent Staining

Sim et al. developed a probe coupling fluorescein and the bisphosphonate alendronate. In comparison with different calcium salts, the probe preferentially binds to hydroxyapatite and shows higher specificity and signal than alizarin red staining, enabling visualization of micro-calcifications [121]. The application was possible both ex vivo and in vitro. Several other dyes targeting hydroxyapatite have been reported, including, for example, the long-established dye calcein [122,123], fluorescence-labeled osteocalcin [124] and an arylphosphonic acid-based dye [125]. The development of new fluorescent dyes may offer several advantages over von Kossa and alizarin red staining. A higher sensitivity and the opportunity for multiplexing are possible by likewise following a very similar protocol. Traditional workflows combined with fluorescent dyes still include tissue sectioning, thus prohibiting a further downstream analysis of tissues. 

##### Near-Infrared Fluorescence Tracer (NIRF)

The NIRF calcium tracer is another dye for vascular calcification. This method is based on fluorescence optical imaging that uses excitation light from the near-infrared spectrum to stimulate fluorophores [126]. Based on the work of Zaheer et al. [127], Osteosense^®^ (PerkinElmer, Boston, MA, USA) is a NIRF calcium tracer consisting of a fluorophore and a bisphosphonate (pamidronate) that binds with great affinity to hydroxyapatite in mineral deposits. This dye is commercially available with fluorophores, exciting approximately at 680, 750 and 800 nm. In human samples and preclinical mouse models, this method has been demonstrated to show early microcalcifications [128]. As they bind to nanocrystals [129], very early lesions can be identified. Osteosense^®^ could discriminate the influence of a Bmp inhibitor on aortic mineralization in a d Ldlr^−/−^ mice model via ex vivo microscopy [130]. In addition, it is applied in several in vivo and ex vivo IVM studies [122,129].

For imaging of NIRF calcium tracers, special microscope equipment is required, depending on the dye employed, including, for example, suitable light sources providing light in the NIR range, cameras with a good response to NIR wavelengths and functioning objectives. Nevertheless, depending on the fluorophore coupled, imaging of Osteosense^®^ can also be possible with an appropriately equipped standard fluorescence microscope. 

In conclusion, histological analysis with the classical dyes alizarin red and von Kossa, imageable in the transmitted light microscope, remains the gold standard with fluorescence dyes offering a valuable alternative and with dye-dependent higher specificity towards hydroxyapatite, which comes with increased technical requirements and higher experimental costs. The calcification area can be localized (intimal/medial) and, depending on the sensitivity of the method, a discrimination between micro- and macrocalcifiation is also possible. Sectioning, staining and imaging of substantial tissue sections is required for quantification and location of calcification, thus resulting in tissue destruction. The requirement of animal sacrifice for histological analysis impedes longitudinal analysis. However, as shown by an elegant experimental design by Hutcheson et al., with an intravenous application of alizarin red and calcein at different time points prior to animal sacrifice and post mortem histological analysis [122], an implementation of a longitudinal approach employing classical histology seems possible.
Figure 1Characteristics of methods of classical microscopy employed for the detection of vascular calcification in mice. Key aspects for the critical appraisal of the applicability of classical histology are summarized with regard to the applicability of classical histology in vivo and ex vivo, the identifiable location and size of vascular calcification, the repeatability of analysis in longitudinal studies, the compatibility with additional methods, and the personal and technical expanse. The exemplary workflow presented is based on the publications of [116,117,118,121,122].
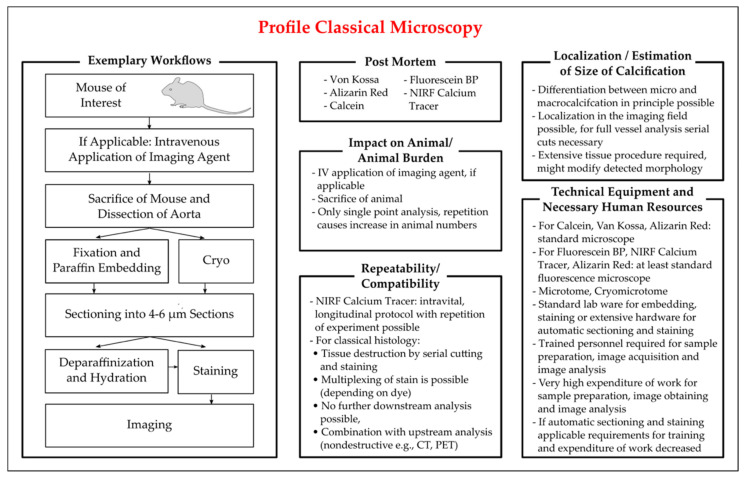


##### Intravital Microscopy 

IVM imaging is carried out in a living species and allows imaging in a physiological in vivo setting. IVM is predominantly used with the application of fluorescent dyes. The application route of the dye depends on the experimental model, for example via intravenous injection. As an alternative to dyes, the application of genetically altered animal models with the expression of a fluorescence reporter is also possible. 

The selected procedures for IVM are summarized in Figure 2. Before starting IVM, the effectiveness of a sound anesthesia and a firm restrainment must be tested to prevent tissue movement. Depending on the time frame of IVM and on the tissue imaged, a prolonged surgery time have to be conducted, which can include, but is not limited to, ventilation, thermal control, infusions, and maintenance or intensification of anesthesia. The desired tissue is isolated, accessed by surgery and imaged. The microscopic options include two-photon microscopy (TPM, also known as laser scanning microscopy or multiphoton microscopy) and confocal microscopy. In TPM, the fluorophore is excited by two or more photons of lower energy levels that have to be absorbed by the fluorophore simultaneously. This offers several advantages, as the lower energy levels of photons permit a deeper tissue penetration and results in less out-of-plane light, thus reducing background and improving the signal-to-noise ratio [131]. TPM also causes less phototoxicity and photobleaching than confocal microscopy, where a single photon at a specific wavelength excites the fluorophore [131]. 

Up to now, most of the experimental procedures using IVM work with an ApoE^−/−^ mouse model and analyze plaque progression with or without a treatment procedure. For studying skeletal development and atherosclerosis, a bone tracer was conjugated using the bisphosphonate pamidronate to the NIR fluorophore IRDye78 [127]. Apart from commercially available Osteosense^®^, an application of other dyes discussed earlier in this article seems possible, e.g., for a fluorescein-coupled alendronate dye recently published by Sim et al., where the authors predicted the potential for a future in vivo application [121]. 

A study by Aikawa et al. combined IVM with µCT to study inflammation and calcification [129]. IVM was conducted repeatedly in mice aged 20 and 30 weeks on an atherogenic diet. They applied bisphosphonate-derived dyes to detect calcification (Osteosense^®^), iron oxide fluorescent nanoparticles detecting macrophage accumulations and an imaging agent for cathepsin K activated upon enzymatic cleavage. That proof-of-principle study clearly highlighted the potential for applying repeated read-out techniques in a longitudinal study in mice [129]. 

Following the principle of those established protocols [129], several other experimental procedures were conducted. In a further study on valve calcification by the same group, IVM was followed by euthanasia, subsequent ex vivo imaging and further histological analysis [132]. To assess the effect of cathepsin S inhibition, mice received coinjection of a protease activatable dye (ProSense^®^) and a bone tracer (Osteosense^®^), followed by fluorescence microscopy both in vivo and ex vivo after dissection of the aorta [133]. With Osteosense^®^, the authors found increased vascular calcification in ApoE^−/−^ mice with CKD compared to ApoE^−/−^ mice with normal kidney function [133]. A longitudinal approach was followed in another protocol. The first IVM was conducted at the beginning of the diet switch, followed by an additional IVM 10 weeks later, which depicted a significant increase in calcification over the prolonged continuation of atherogenic diet [122]. 

Apart from cardiovascular calcifications, Köppert et al. published an interesting protocol on the clearance of CPP [81]. They created fluorescent CPP particles by labeling bovine fetuin-A with the fluorescent dyes Alexa 488 or Alexa 546, which were then used to prepare primary and secondary CPP. Mice were soundly anesthetized, the liver was extraventralized and the mouse was mounted on a thermostatic microscope with the liver downward. Mice were injected with Hoechst 33258, tetramethylrhodamine ethylester, primary and secondary CPP, and intravital TPM was continuously recorded. The authors found liver sinusoidal endothelial cells to clear primary CPP, whereas liver Kupffer cells preferentially cleared secondary CPP [81]. 

In summary, IVM requires extensive and in part customized technical equipment as well as extensive researcher training. However, IVM, especially TPM, offers an excellent resolution for studying calcification and underlying pathophysiological processes in vivo, providing researchers with images in the biological setting. IVM can enable longitudinal study design [122]. After image acquisition, post-operational care by suturing of the wounds and post-operative analgesia is necessary. Euthanasia of the animal at the end of IVM and tissue dissection allows further post mortem analysis. Under the limitation of good tissue preservation, further analysis of the tissue in downstream analytical processes is possible, considering that the applied fluorophore does not interact with the downstream analysis procedure.
Figure 2Characteristics of protocols of intravital microscopy (IVM) employed for the detection of vascular calcification in mice. Key aspects for the critical appraisal of the applicability of IVM are summarized with regard to the applicability of IVM in vivo and ex vivo, the location and size of vascular calcification identifiable by IVM, the repeatability of the read out in longitudinal studies, the compatibility with additional methods, and the personal and technical expanse. The exemplary workflow presented is based on the publications of [81,127,129,130,132,133].
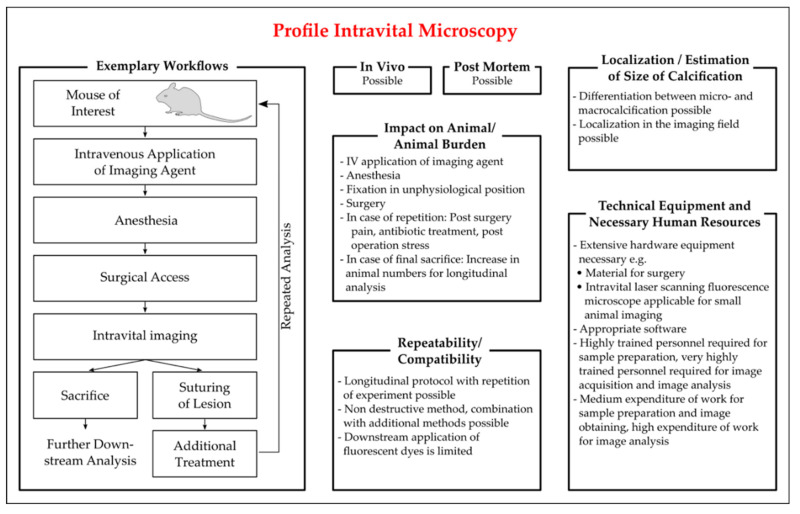


#### 3.3.3. Magnetic Resonance Imaging

MRI is a promising in vivo imaging modality for animal models. It is a noninvasive imaging method that uses strong magnetic fields and high-frequency radio waves to produce detailed images of the tissue structures [134]. By using contrast agents like nanoparticles, superparamagnetic iron oxide or gadolinium particles, MRI, combined with PET, can provide information on cellular and molecular targets of endothelium dysfunction, mineral deposition, and inflammation [135]. MRI alone, without a contrast agent targeting calcium, may not be sufficient to identify vascular calcification. 

#### 3.3.4. X-ray

Mineralized tissue highly attenuates X-rays. Therefore, CT allows the visualization of calcification foci. For humans, non-invasive CT is a well-established tool to detect coronary artery calcium (CAC) [136]. Due to developments increasing the spatial resolution capabilities of µCT, CT becomes increasingly applicable for small rodent models, predominantly depicting macrocalcifications, but ongoing improvement in spatial resolutions renders the robust detection of microcalcification possible. So far, the µCT analysis for rodent models has been used post mortem for whole animals or explanted tissue as well as in in vivo settings. 

A short overview of the experimental procedure is given in Figure 3. Although for µCT both in vivo and post mortem analysis of calcification is possible, in vivo analysis is hampered by physiologic tissue movement and limited achievable resolution. For in vivo µCT, sound animal anesthesia is necessary, both for animal welfare and control of tissue movement. If the analysis is carried out post mortem, the vessels can be analyzed after dissection and fixation or an analysis is also possible for whole animal preparations, without dissection of the tissue. To enhance contrast, application of contrast agents can be necessary and for optimization of image acquisition, sample immobilization can be required. Both contrast agents and sample immobilization can impede further downstream analysis.

A study analyzing embedded murine aortas from Ldlr^−/−^ mice at different chows detected increased vessel calcification via µCT [74]. The downstream histological analysis with alizarin red stain confirmed the CT-detected calcium volume [74]. Similar correlations between alizarin red-stained calcium foci and corresponding µCT images were found in Enpp^−/−^ dissected aortas [137]. This protocol optimized the soft tissue definition by formalin fixation of the aorta, immersion in macro-molecular iopamidol-based contrast agent, and filling and submersion of the aorta in corn oil before CT imaging [137]. Others perfused the animals with PBS and paraformaldehyde prior to organ dissection, followed by formalin fixation, application of sodium azide, and a CT scan in corn oil, confirming CT-identified calcifications with von Kossa staining [138]. Post mortem scans of the heart, aorta and kidney from Enpp^asj/asj^ mice revealed a visible calcification of two-thirds of the animals, while subsequent histological staining showed dramatic calcification in all animals [57].

Besides the post mortem application for rodents, the CT scan is also used for in vivo detection of calcification foci. However, the µCT for rodents is still limited in detecting microcalcifications. Thus, a combination with other imaging technologies, like the use of a radiotracer or in vivo/ex vivo microscopic analysis, is often used. A study by Aikawa et al. presented the earlier-combined µCT imaging with IVM [129]. The study aimed to depict the involvement of inflammation in the process of vascular calcification by likewise demonstrating the potential of these technologies in mice and found IVM-visualized osteogenic signals were undetectable by µCT [129]. A longitudinal approach of CT scans in rodents was used in an adenine-induced chronic renal failure model in rats [26]. This study performed longitudinal feasibility and reproducibility studies. The animals were scanned repeatedly in pentobarbital anesthesia within one (reproducibility) or two (feasibility) weeks. A comparison between in vivo µCT to ex vivo µCT found that in vivo analysis was hampered by movement artefacts, although heavily calcified aortas were, due to stiffness, less affected by movement artefacts than only slightly calcified aortas [26]. Nevertheless, a comparison between in vivo and ex vivo results was not possible due to post mortem distortion [26].

For further and more detailed information on µCT utilization for detection of vascular calcification in animal models, the reader is referred to a recent review [139]. Currently, the achievable resolution of µCT is not sufficient to robustly detect microcalcification in small rodent models in vivo. To overcome this problem, next to a further improvement of µCT resolution, the combination of µCT with PET has the potential to enable researchers to identify and differentiate micro- and macrocalcification in vivo.

In summary, µCT already offers a valuable tool for detecting vascular calcification both post mortem and, to a lesser extent, in vivo. A longitudinal study is in principle possible, thus helping researchers to reduce deviation and help to prove the statistical significance, even in smaller sample sizes, by using each animal as its control by likewise facilitating the search for the optimum treatment period for analytical procedures and for a follow-up study. 

Due to the required hardware equipment and personnel training required, µCT is a technique that will be primarily available in specialized core facilities.
Figure 3Characteristics of protocols of Micro-CT (µCT) employed for the detection of vascular calcification in mice. Key aspects for the critical appraisal of the applicability of µCT are summarized with regards to the applicability of µCT in vivo and ex vivo, the location and size of vascular calcification identifiable by µCT, the repeatability of read out in longitudinal studies, the compatibility with additional methods and the personal and technical expanse. The exemplary workflow presented is based on the publications of [26,74,129,137].
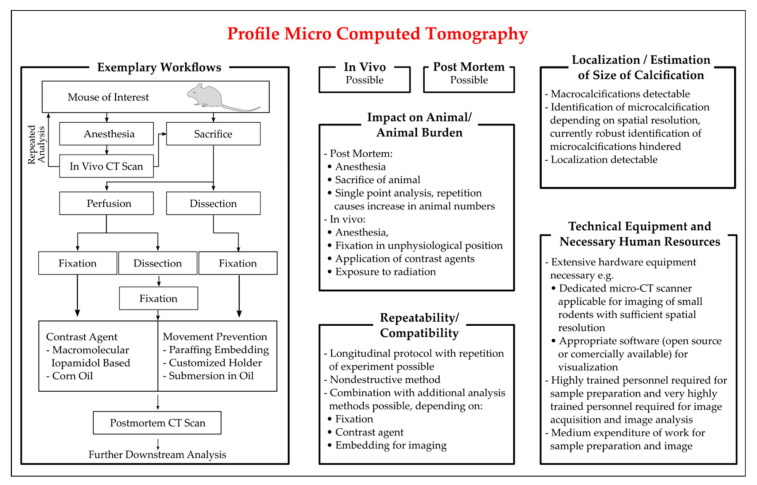


#### 3.3.5. Positron Emission Tomography

To image target structures, PET requires a radiotracer that emits the rays detected in PET. The radiotracer utilized for the detection of vascular calcification in PET scans is ^18^F-NaF. ^18^F-NaF binds specifically and sensitively to calcium deposits [140]. Due to the higher surface of microcalcifications, ^18^F-NaF preferentially binds to them and can be employed to differentiate between macro- and microcalcifications when used in combination with CT [136]. A short experimental overview of PET is given in Figure 4. PET scans can be conducted in vivo as well as post mortem. For in vivo analysis, animals are anesthetized and injected with a radiotracer and, if applicable, a contrast agent for CT acquisition. After CT acquisition for attenuation, in vivo PET scanning is conducted. Although longitudinal studies with repeated PET scans are possible, the majority of animals are sacrificed after the PET scan for subsequent histological analysis and comparison of the results. For imaging and quantification of calcification, the establishment of reconstruction parameters is decisive. A study evaluating PET performance found a significant impact on the assessment of the mineralization process by optimizing the reconstitution parameters in uremic and non-uremic rats [141]. In ApoE^−/−^ mice on a high cholesterol diet, a ^18^F-NaF PET/CT scan was conducted in vivo in isoflurane-anaesthetized mice with an intraperitoneal tracer injection, as well as ex vivo and calcification were found to progress over the time of high cholesterol feeding with results consistent between in vivo and ex vivo settings [142]. To assess the effects of vitamin K and warfarin on calcification, a recent study analyzed ApoE^−/−^ mice on different treatment regimens with ^18^F-NaF PET/CT and a subsequent histological analysis [143]. While dense calcifications detectable in the warfarin group corresponded to the respective von Kossa staining, the authors found PET results to discriminate between early- and advanced-stage plaques [143]. A longitudinal approach with two PET scans, before and after animal exercise sessions lasting nine weeks, was conducted by Hsu et al. [144]. They found the microarchitecture of calcium deposits to differ between mice at different exercise levels. Calcification identified with ^18^F-NaF PET was consistent with the results derived from alizarin red staining [144].

If a PET scan is carried out post mortem, the radiotracer can be injected into the living animal and the tissue is dissected, perfused and fixed after animal sacrifice, followed by CT and PET acquisition. Ex vivo PET scans monitored the ongoing calcification progress after 12, 20 and 30 weeks of investigation and found increasing ^18^F-NaF uptake over time [145]. One hour after the tracer injection, the mice were sacrificed and aortas were isolated, fixed, and subsequently imaged with the PET [145]. The scans were flanked by detecting biochemical markers and subsequent histological staining and found alizarin red and HE staining to confirm the results derived from ^18^F-NaF PET scans [145].

In conclusion, PET is a promising technique for the detection of vascular calcification. After the recent publication of proof-of-principle studies, further auspicious protocols for the application of PET for animal models of vascular calcification are expected, permitting also the application of longitudinal studies. As for µCT, MRI and IVM, upscale hardware equipment and comprehensive training of personnel will focus on PET in specialized core facilities.
Figure 4Characteristics of protocols of positron emission tomography (PET) employed for the detection of vascular calcification in mice. Key aspects for the critical appraisal of the applicability of PET are summarized with regards to the applicability of PET in vivo and ex vivo, the location and size of VC identifiable by PET, the repeatability of analysis in longitudinal studies, the compatibility with additional methods, and the personal and technical expanse. The exemplary workflow presented is based on the publications of [140,141,142,143,144,145].
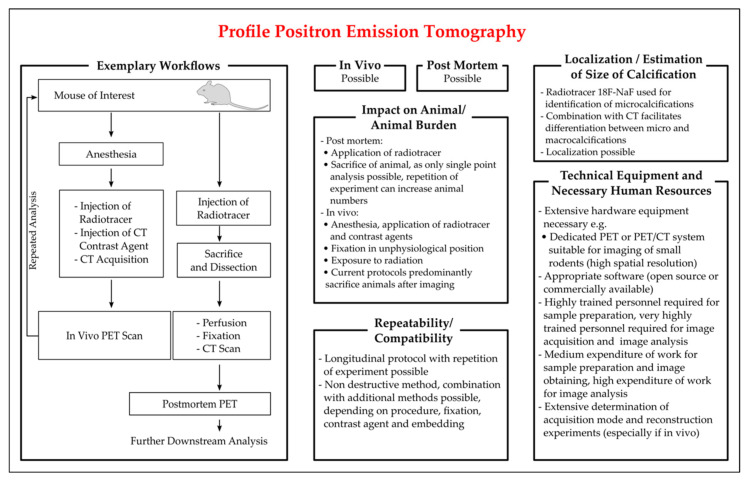


## 4. Discussion and Conclusions

Technical progress, especially the increase in resolution, enabled a technological transfer from bed to benchside and provided researchers with many new technological methods applicable for the identification of vascular calcification in small rodent models. Historically, direct identification and assessment of vascular calcification were only possible ex vivo. Technologies like the o-cresolphthalein assay and histological staining require tissue dissection and less invasive methods, permitting repetitive assessment, were only available for surrogate parameters like serum markers of calcification. Now, new state-of-the-art, non- to low-invasive technologies have become increasingly available, making the repetitive assessment of vascular calcification in rodent models achievable. This development is promising for researchers and animal welfare alike, as the utilization of these technologies offers researchers the opportunity to assess vascular calcification in animals repetitively and in a much shorter time. Researchers can gain more versatile data by likewise having the chance to decrease the animal numbers and animal burden and therefore have the opportunity to translate “Reduce and Refine” from 3R theory into practice.

## Figures and Tables

**Table 2 biology-10-00459-t002:** Selected methods for the detection of vascular calcification in rodents. Methods where samples are subject to intense chemical or physical strain and/or sample restoration is not possible are rated as destructive methods. Methods which administer a light strain, or when sample restoration is possible, are termed non-destructive.

Items		Multi Point Analysis Feasible	Single Point Analysis	
			Destructive	Non-Destructive
Biochemical markers of pathological calcification		Circulating blood markers	Detection of resident markers in tissue	-
Markers of vessel stiffness		Pulse wave velocity	Wire myography	-
		Pulse pressure		-
Direct detection of calcification	Biochemical	-	*o*-cresolpthalein	-
			Alizarin red	
	Microscopy	Intravital microscopy	Histological staining of dissected tissue	Intravital microscopy
	Magnetic Resonance	Magnetic resonance imaging	-	Magnetic resonance imaging
	X-ray	(Micro-) Computed tomography	-	(Micro-) Computed tomography
	Positron emission	Positron emission tomography	-	Positron emission tomography

**Table 3 biology-10-00459-t003:** Selected serum biomarkers for vascular calcification utilized in rodent models [2,82].

Biomarker	Mechanism	Consequences in Genetically Altered Mice/Correlation with Endpoint	Methods of Biomarker Analysis
Calcium [78]	Causal relationship between elevated calcium and calcification unclear	In humans, correlation of high serum calcium with coronary atherosclerosis, cardiovascular events and increased mortality.	Standard serum chemistry
Phosphate [78,83]	Pathophysiologic mechanisms between elevated phosphate and calcification are incompletely understood, several pathways likely contribute to increased mortality in ESRD patients	In humans, increased serum phosphate correlate with increased coronary calcification, morbidity and mortality.	Standard serum chemistry
Alkaline Phosphatase [84]	Hydrolyzation of extracellular pyrophosphate and formation of hydroxyapatite	Alkaline phosphatase is increased in models with medial calcification.	Standard serum chemistry/Functional assay, Antibody-based technique
Calcium Propensity [85,86]	Physiologically, formation of calciprotein particles (CPP) is tightly regulated in serum supersaturated in calcium and phosphate by inhibitors such as Fetuin-A	In human studies, correlation with all-cause mortality in predialysis patients.	In vitro test that monitors the maturation time of calciprotein particles via nephelometry
Matrix Gla Protein [67]	Inhibitor of mineralization Binds hydroxyapatite crystals Inhibitor of Bmp-2	Mgp^−/−^ mice exhibit extensive mineralization of the aorta located predominantly in the media.	Antibody-based technique
Fetuin-A [68,69,81]	Inhibitor of mineralization Binds to calcium and phosphate and forms inactive complexes	Mice deficient of Fetuin-A develop soft tissue calcifications, with extent depending on genetic background and chow.	Antibody-based technique
Osteopontin [68]	Inhibitor of mineralization Inhibits hydroxyapatite formation Activates osteoclast function	Knockout of Opn alone does not induce calcification in mice. Double Knockout of Opn and Mgp in mice exacerbates vascular calcification in comparison to sole knockout of Mgp.	Antibody-based technique
Fgf-23 [49]	Regulates phosphate homeostasis and metabolism of Vitamin D, binding of Fgf-23 to receptor requires Klotho as necessary co-factor	Fgf-23^−/−^ exhibit early onset vascular calcification. Elevated Fgf-23 levels associated with higher calcification scores and arterial stiffness.	Antibody-based technique
Klotho [46]	Inhibitor of mineralization Activation of Fgf-23 Inhibition of VSMC osteoblastic transdifferentiation	Klotho^−/−^ mice exhibit severe vascular calcification.	Antibody-based technique
Pyrophosphate (PPi) [52,54,55,56]	Inhibitor of mineralization Prevents the nucleation of amorphous calcium phosphate Prevents crystallization and crystal growth of hydroxyapatite	Enpp^−/−^, Enpp^ttw/ttw^ and Enpp^asj^ exhibit vascular calcification of the aorta. Abcc6^−/−^ mice exhibit increased arterial calcium content.	Enzymatic assay

Abbreviations: Abcc6—ATP binding cassette subfamily C member 6; asj –ages with stiffened joints; Enpp—ectonucleotide pyrophosphatase phosphodiesterase; ESRD—end-stage renal disease; Fgf-23—fibroblast growth factor 23; Mgp—matrix gla protein; Opn—osteopontin; Pyrophosphate—PPi; ttw—tip toe walking.

## Data Availability

Not applicable.

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
