# Peer review of "Vascular Calcification in Rodent Models—Keeping Track with an Extented Method Assortment"

_biology, 2021, doi:10.3390/biology10060459_

Round 1

Reviewer 1 Report

The manuscript is a narrative review of the different techniques available to analyze vascular calcification in rodent models, the strengths and weaknesses of these techniques, and their applicability against the 3Rs.

All sections and figure legends are written without inconsistencies and in a logical and coherent form. The topic is of great interest for researchers.

This descriptive review needs minor improvements: 

  • Please double check spellings throughout the manuscript. For example, page 1, paragraph 2, you say ‘extend’ instead of ‘extent’. ‘Extend’ is also used elsewhere in the manuscript e.g. page 3.

Page 8, line 302 and page 10, line 397 you say ‘repeadetly’ instead of ‘repeatedly’.

Page 7, line 247: there is an extra ‘d’ before LDLR-/-.

  • The name Alizarin Red should go in lowercase throughout the whole text: alizarin red.

  • The name o-cresolphthalein should have the ‘o’ in italics: o-cresolphthalein.

  • Table 1: Please check consistent use of acronyms throughout the table (e.g. Opg-/- or OPG-/- ; Abcc6-/- or Abcc-/-).

  • Table 1, Lldlr-/- : did you mean 'Ldlr-/-' ? On page 7, line 247, you also say ‘LDRL-/-‘. Please be consistent with your use of upper or lowercase.

  • Table 1: Please be more specific when you mention ‘physical impairment’?

  • Table 1, Mgp-/- : ‘…..heart beats increase…’. Do you mean heart rate increases?

  • Section 1, Introduction, paragraph 1, lines 5-6: The use of ‘lesion localization’ and ‘lesion size’ could be suggestive of an atherosclerotic plaque, which is not the case here. I would simply suggest using ‘calcification localization’ and ‘calcification size’.

  • Section 2, Rodent models for induction of vascular calcification: It would be interesting to expand on the use of in vitro and ex vivo models in vascular calcification. Whilst they do offer a valuable alternative to animal models, they are of course limited – particularly in the context of a CKD or atherosclerotic background. It might be helpful for the reader to comment on this, and to confirm the importance of using animal models in vascular calcification research.

  • 2.2. Ankle-brachial index: has this measurement actually been used in rodent models of calcification? If not, do you think this measurement is feasible/could be easily applied to rodent models?

  • 3.2 Histological staining for calcium deposits: it might be helpful to discuss the other disadvantages associated with the sectioning of calcified tissue e.g. calcification dropping out during sectioning, the number of tissue sections that should be analyzed from a single blood vessel to avoid bias.

  • You briefly discuss the ability of the different techniques to differentiate between micro- and macrocalcifications. Could you also discuss their ability to differentiate between intimal and medial calcification? This was not clear in the text. 

Reviewer 2 Report

Thank you for asking me to review the article “ Vascular Calcification in Rodent Models – _Keeping Track with an Extended Method Assortment  article on methods for examining vascular calcification.” The article was clear, well written and informative.

As a review article on techniques for use in animal models think that it was relatively comprehensive, although at times (perhaps necessarily) a little bit oversimplified, and nicely follows from the previous article by the same authors on animal models.

I have a few minor comments;

  1. Functional vessel stiffness is not just a function of calcification. There are a number of matrix proteins involved in the stiffening process, not to mention cellular responses, and elastin appears important also. It is important to stress this in the discussion of PWV/ABI/PP. There are also endothelial systems (esp related to NOS) that may play a role in vascular stiffness ( as well as VSMC and calcification components) that are unrelated to vascular calcification.

  1. It is important to define how increases in PP may be driven by changes in the timing of reflected waves and arterial stiffness, usually due to increase in SBP rather than reduction in DBP. The former may not have much to do with vascular calcification.

  1. In the PET section, I was not aware that PET techniques can be applied post-mortem and I am not sure I understand how this could be done anyway. If such a technique is available, perhaps the authors could expand on how this is possible as it is not readily apparent.

  1. I am unsure if figure 1 adds much with the intended audience likely aware of these issues.

  1. Minor issues

3R principle explain in abstract – reduce reuse recycle

ESRD not defines in abbreviations table 3
